# Growth Response of Non-Conventional Yeasts on Sugar-Rich Media: Part 2: Citric Acid Production and Circular-Oriented Valorization of Glucose-Enriched Olive Mill Wastewaters Using Novel *Yarrowia lipolytica* Strains

**DOI:** 10.3390/microorganisms11092243

**Published:** 2023-09-06

**Authors:** Dimitris Sarris, Erminta Tsouko, Angelos Photiades, Sidoine Sadjeu Tchakouteu, Panagiota Diamantopoulou, Seraphim Papanikolaou

**Affiliations:** 1Laboratory of Physico-Chemical and Biotechnological Valorization of Food By-Products, Department of Food, Science and Nutrition, School of the Environment, University of the Aegean, Leoforos Dimokratias 66, 81400 Myrina, Lemnos, Greece; etsouko@aegean.gr (E.T.); anp61@aber.ac.uk (A.P.); 2Laboratory of Food Microbiology and Biotechnology, Department of Food Science and Human Nutrition, Agricultural University of Athens, Iera Odos 75, 11855 Athens, Attiki, Greece; sadjeu2000@yahoo.fr; 3Institute of Technology of Agricultural Products (ITAP), Hellenic Agricultural Organization—Demeter, 1 Sofokli Venizelou Street, 14123 Lykovryssi, Attiki, Greece; pdiamantopoulou@elgo.gr

**Keywords:** sustainable bioprocessing, bioactive compounds, microbial oil, polysaccharides, citric acid

## Abstract

The global market for citric acid (CA) is one of the biggest and fastest expanding markets in the food industry. The CA production employing microbial bioprocessing with efficient GRAS strains and renewable waste streams is in line with the European Union binding targets for resource efficiency, sustainable consumption-production, and low-carbon technologies. In this work, the potential of three novel wild-type *Yarrowia lipolytica* strains (namely LMBF Y-46, LMBF Y-47 and ACA-YC 5033) regarding the production of CA and other valuable metabolites was tested on glucose-based media, and the most promising amongst the screened strains (*viz*. the strain ACA-YC 5033) was cultured on glucose-based media, in which part of the fermentation water had been replaced by olive-mill wastewaters (OMWs) in a novel approach of simultaneous OMW valorization and bioremediation. In the first part of this study, the mentioned strains were cultured under nitrogen-limited conditions with commercial (low-cost) glucose employed as a sole carbon source in shake-flask cultures at an initial concentration (S_0_) ≈ of 50 g/L. Variable quantities of secreted citric acid (CA) and intra-cellular compounds (*viz*. polysaccharides and lipids) were produced. All strains did not accumulate significantly high lipid quantities (i.e., maximum lipid in dry cell weight [DCW] values ≈30% *w*/*w* were noted) but produced variable CA quantities. The most promising strain, namely ACA-YC 5033, produced CA up to *c*. 24 g/L, with a yield of CA produced on glucose consumed (Y_CA/S_) ≈ 0.45 g/g. This strain in stirred tank bioreactor experiments, at remarkably higher S_0_ concentrations (≈110 g/L) and the same initial nitrogen quantity added into the medium, produced notably higher CA quantities, up to 57 g/L (Y_CA/S_ ≈ 0.52 g/g). The potential of the same strain (ACA-YC 5033) to bioremediate OMWs and to produce value-added compounds, i.e., yeast cells, CA, and intra-cellular metabolites, was also assessed; under nitrogen-limited conditions in which OMWs had partially replaced tap water and significant glucose concentrations had been added (S_0_ ≈ 100 g/L, simultaneous molar ratio C/N ≈ 285 g/g, initial phenolic compounds [Phen_0_] adjusted to ≈1.0 g/L; these media were similar to the OMWs generated from the traditional press extraction systems) the notable CA quantity of 60.2 g/L with simultaneous Y_CA/S_ = 0.66 g/g, was obtained in shake flasks, together with satisfactory phenolic compounds removal (up to 19.5% *w*/*w*) and waste decolorization (up to 47.0%). Carbon-limited conditions with Phen_0_ ≈ 1.0 g/L favored the production of yeast DCW (up to 25.3 g/L), with equally simultaneous interesting phenolic compounds and color removal. The fatty acid profile showed that cellular lipids were highly unsaturated with oleic, linoleic and palmitoleic acids, accounting for more than 80% *w*/*w*. This study proposed an interesting approach that could efficiently address the biotreatment of toxic effluents and further convert them into circular-oriented bioproducts.

## 1. Introduction

The agricultural sector produces 23.7 million t of food globally on a daily basis, while the growing world’s population is projected to raise requirements for food production by 1.6 folds until 2050. This will inevitably lead to the enormous generation of agri-food waste streams, completely diverging from the goals of the “2030 Sustainable Development” Agenda (SDG) [1] that require resource efficiency in consumption and production, decoupling of economic growth from environmental degradation and sustainable consumption and production patterns. In addition, the Whole-of-Economy approach of the “Fit for 55” focuses on the overall reduction of emissions, the EU’s green recovery from COVID-19, and the modernization of relevant sectors. The package of “Industrial Transformation and Carbon Pricing” sets bindings for the industry to decarbonize production processes and to promote innovative, low-carbon technologies and green jobs while the EU is proposing that net greenhouse gas emissions be reduced by at least 55% by 2030 compared to 2005 [2].

Being in line with the aforementioned, agri-food residues that are abundant, low- (or negative-) cost and sustainable should be valued as resources and re-introduced in value chains of the circular economy. This could be achieved via the implementation of microbial bioprocessing [3,4]. The fermentation media is critical for the efficient biosynthesis of microbial metabolites such as citric acid (CA) since it corresponds to more than 50% of the total production cost. CA is an organic tricarboxylic acid with many applications due to its GRAS (“Generally recognized as safe”) nature, high water solubility and enhanced chelating and buffering potential. CA is normally used as an emulsifier, stabilizer, flavoring agent, and texturizer in the food sector, while its use in pharmaceuticals and agriculture has also been reported [5]. The outbreak of the COVID-19 pandemic led to supply deficiencies, with consumers stocking up and preserving numerous food supplies, with CA coming in the foreground. The global CA market will be valued at USD 9.4 billion in 2029 at a compound annual growth rate of 5.9% within the forecast period of 2022–2029 [6].

CA production has been widely investigated as metabolite generated through fermentations performed when several renewable resources, including beet and cane molasses, glycerol, corn starch, brewery effluents, lignocellulosic-based hydrolysates and oils from soybean, rapeseed, palm and olive [7] were used as microbial substrates. The most established microorganism for CA production is the fungus *Aspergillus niger* [8], while complications related to the strain’s opportunistic pathogenic and allergic nature have led researchers seeking for alternative strains [9]. *Yarrowia lipolytica* is considered as a yeast with great potential for biorefinery development. It has the ability to metabolize diverse fermentation substrates, including C5 and C6 sugars, cellulose- and hemicellulose-based hydrolysates derived from lignocellulosic biomass, biodiesel-derived glycerol, and fatty materials [9,10], and produce various quantities of CA from growth on several of the mentioned substrates [5,9,10,11,12]. In addition to CA, *Y. lipolytica* can produce intra-cellular polysaccharides (IPs) and lipids of high unsaturation degree, simultaneously offering enhanced decolorization and phenolic removal of wastewaters, i.e., olive-mill wastewaters (OMWs), which have a heavy organic load [11]. In terms of the production of CA by *Y. lipolytica* yeasts, by far, the most important parameter governing the success of the process is the utilization of wild-type or mutant/genetically modified strains (mostly acetate and/or acotinase mutants) [12]. For instance, in most cases, the CA_max_ quantity produced for the wild-type strains is some tenths g/L (i.e., a concentration up to 60 g/L is considered as very satisfactory [12]), whereas by mutant or genetically engineered strains, the corresponding concentrations can be in some cases ≥100 g/L [12]. The maximum concentration of CA produced so far (≈200 g/L, with conversion yield on sugar consumed =0.85 g/g and productivity = 0.51 g/L/h) has been reported when a genetically engineered strain of *Y. lipolytica* was cultivated on renewable inulin [13].

Waste streams that contain high moisture levels or occur as wastewater are prone to contamination, while their storage is difficult due to high space requirements [14]. Such a substrate is olive mill wastewater (OMW). Olive oil is mainly produced within the Mediterranean territory that covers almost 97% of the worldwide production. The most applied systems for olive oil extraction include the three-phase decantation and the two-phase centrifugation [15]. Huge amounts of OMWs, equal to 30 million m^3^/per year, are produced globally involving these extraction processes [16,17]. The organic matter of OMWs, i.e., residual oil, phenolic compounds, organic acids, C5 and C6 sugars (mostly glucose) and several nitrogenous molecules, highly depends on the extraction process [18,19,20,21]. The common practice of disposing of OMWs in soil and waterways creates severe environmental impact and ecosystem imbalances due to their high content of phenolic compounds, sugars, fatty acids, and tannins [19]. Therefore, their management via eco-friendly methods and biotechnological conversion routes with efficient microbial strains to obtain value-added products is a one-way option, especially since their integration within biorefinery strategies is state-of-the-art and emerging [14].

Recent developments in the research related to the biotechnological valorization of OMWs indicate that these liquid streams should be considered as a fermentation medium to valorize rather than a waste to discharge, being as both a process water and a potential liquid substrate for various fermentation processes [11,20,21,22,23]. In several cases, OMWs have been employed to “dilute” “concentrated” residues and waste streams like molasses and crude glycerol [14,24,25]. It is evident that a prerequisite for the successful accomplishment of the mentioned bioprocess would be that the implicated microorganisms would withstand the (potentially elevated concentrations of the) phenolic compounds found in the OMW-based media [14,18,20,21,23,24,26] while of significant interest, would be the simultaneous detoxification (*viz*. dephenolization and decolorization) of the OMW-based media, together with the production of added-value metabolites [14,18,23,24,26].

The present study was initially related to the evaluation of the capacity of three “novel” strains of the species *Y. lipolytica* that had not been extensively studied in connection to their growth response on glucose-based media. These strains, namely LMBF Y-46, LMBF Y-47 and ACA-YC 5033, were cultured under nitrogen-limited conditions employed in order to “boost” both the production of CA and the accumulation of cellular lipids [12,27] with commercial (low-cost) glucose employed as a sole carbon source in shake-flask experiments. The most promising among the tested microorganisms regarding CA production, namely *Y. lipolytica* ACA-YC 5033, was further cultured in laboratory-scale bioreactor experiments to further boost the production of the mentioned organic acid. Thereafter, this strain was grown on glucose-supplemented OMW, employing shake flask fermentations. Batch experiments involved OMW supplementation with glucose, at different C/N ratios, different initial phenolic compounds concentrations (Phen_0_) and different (moderate or elevated) initial concentrations (S_0_) of glucose (i.e., S_0_ ranging between *c*. 40 and *c*. 100 g/L). It is noted that moderate S_0_ concentrations (i.e., ≈40 g/L) can be found in OMWs deriving from three-phase centrifugation systems with low water consumption [21] (the so-called “2.5-phase” centrifugation systems) while high S_0_ concentrations (i.e., S_0_ ≥ 80 g/L) can be found in the traditional (*viz*. press extraction) olive oil production systems [18,20,21]. The simultaneous production of yeast biomass, microbial lipids and intra-cellular polysaccharides (IPs) was investigated while the decolorization and the dephenolization of the wastewater and the profile of fatty acids were also monitored. Overall, this study reports the green bioprocessing of OMW to produce circularly oriented and biobased products that contain CA and other co-products with favorable nutritional profiles, including lipids and IPs, with simultaneous partial detoxification (*viz*. decolorization and dephenolization) of the OMWs employed as substrates.

## 2. Materials and Methods

### 2.1. Raw Materials and Microorganisms (Maintenance and Inoculum Preparation)

*Y. lipolytica* LMBF Y-46 and LMBF Y-47 were isolated from gilt-head (sea) bream (*Sparus aurata*) fish [28]. *Y. lipolytica* ACA-YC 5033 was isolated from Greek traditional wheat sourdough [29]. Strains were maintained on potato dextrose agar (PDA) slopes, and Petri dishes at 4 °C. Pre-cultures were prepared in 250 mL Erlenmeyer flasks with 50 mL active volume, containing (in *w*/*v*) 1% yeast extract, 1% bacteriological peptone and 1% glucose. Flasks were incubated at 28 °C, 180 RPM for 48 h in an orbital shaker (New Brunswick Scientific, Edison, NJ, USA).

OMW was provided by an olive oil mill (three-phase) located in Kalentzi (Corinthia, Peloponnese). Prior to its use, it was centrifuged for 15 min at 4 °C, RCF 7690× *g* (Hettich Universal 320R, Vlotho, Germany), and the supernatant was collected and characterized. OMW phenolic content expressed as gallic acid equivalent was 10.0 ± 0.5 g/L. The sugar concentration of the residue was 30.0 ± 0.5 g/L, expressed as glucose equivalent. This waste stream contained negligible quantities of olive oil (0.4 ± 0.1 g/L—determination of oil conducted after triple extraction with hexane). Organic acids were also present in small quantities. The principal organic acids detected were acetic acid (2.0 ± 0.5 g/L) and gluconic acid (2.0 ± 0.5 g/L).

Corn glucose, the main industrial low-value material utilized in confectionary and fermentation industries (≈88% *w*/*w* purity, with impurities being composed of maltose at 6% *w*/*w*, malto-dextrines at 4% *w*/*w*, water at 1% *w*/*w* and salts at 1% *w*/*w*) (Cargill Incorporated, Sheboygan, WI, USA), was used as the main carbon source in the various experiments performed. OMWs were, in several cases and at several concentrations, enriched with the mentioned carbon source.

### 2.2. Batch Shake Flask and Bioreactor Fermentations

In most of the experiments, CA production was carried out in 250-mL Erlenmeyer flasks with 50 ± 1 mL active volume (sterilized growth medium at 121 °C for 15 min). A mineral solution containing (g/L) KH_2_PO_4_ 7.0; Na_2_HPO_4_·2H_2_O, 2.5; MgSO_4_·7H_2_O, 1.5; CaCl_2_·2H_2_O, 0.15; FeCl_3_·6H_2_O, 0.15; ZnSO_4_·7H_2_O, 0.02; MnSO_4_ H_2_O, 0.06 was used. In the first set of experiments (“screening”), trials were carried out on media containing initial glucose (S_0_) at ≈50 g/L, while the nitrogen sources used were peptone (at 1.0 g/L) and yeast extract (at 1.0 g/L). Peptone contained *c*. 18% *w*/*w* nitrogen, whereas the yeast extract contained *c*. 7% *w*/*w* nitrogen. In the first set of experiments, the initial mass ratio C/N was adjusted to ≈80 g/g according to the calculations made. This elevated mass ratio C/N was employed in order to “boost” the production of CA and/or the accumulation of storage lipids [12,27].

In the case of the experiments with OMWs added, properly diluted OMWs (resulting in liquid media containing ≈1.0 g/L and ≈2.0 g/L of initial phenolic compounds concentrations (Phen_0_); these Phen_0_ are amongst the typical ones that can be found in various types of OMWs [11,14,18,21]) were used at the fermentation media. In these experiments, the strain ACA-YC 5033 (selected from the previous screening trials) was used as a cell factory. Initially, two C/N ratios were evaluated for CA production. More specifically, when *c.* 114 g/g C/N initial ratio was applied, the fermentation medium was supplemented with 0.5 g/L (NH_4_)_2_SO_4_ (that contains nitrogen to *c*. 21% *w*/*w*) and 0.5 g/L yeast extract (that, as mentioned, contains nitrogen at *c*. 7% *w*/*w*) and glucose to reach an initial sugar concentration (S_0_) of *c.* 40 g/L. In the case of the *c.* 16 g/g C/N ratio, the medium was supplemented with 4.0 g/L (NH_4_)_2_SO_4_ and 2.0 g/L yeast extract and glucose to reach an S_0_ of around 40 g/L. In each case, a synthetic medium with no addition of OMW was used as a control for comparison. The mentioned S_0_ concentration (≈40 g/L) is typical and can be found in OMWs deriving from three-phase centrifugation systems with low water consumption [21] (*viz*. the 2.5-phase systems). Thereafter, one supplementary shake-flask experiment was also performed under nitrogen-limited conditions (as previously, the fermentation medium was supplemented with 0.5 g/L (NH_4_)_2_SO_4_ and 0.5 g/L yeast extract) in which the S_0_ concentration was adjusted to *c*. 100 g/L (initial C/N ≈ 286 g/g). In this trial, OMWs were added into the medium to yield a Phen_0_ ≈ 1.0 g/L. This trial was chosen since, according to the literature, in several cases (i.e., utilization of the traditional press extraction system), OMWs containing significant quantities of sugars (mostly glucose in concentrations ≥ 80 g/L) may exist [21]. Moreover, the combined increase in the S_0_ concentration and initial C/N ratio may lead to increased CA production, given that the mentioned parameters are prerequisites for substantial production of this organic acid [7,9,12]. Shake flasks were inoculated with 2% *v*/*v* of a 48 h grown pre-culture (see Section 2.1) in an orbital shaker (New Brunswick Scientific, Edison, NJ, USA) (28 °C, 180 RPM). The pH of the medium was kept in a narrow range (pH = 5–6), using 5 M KOH (under aseptic conditions) throughout the fermentation.

One batch fermentation was also conducted with the strain ACA-YC 5033 in a laboratory scale bioreactor (apparatus New Brunswick Scientific Co, Edison, NJ, USA) in media with higher S_0_ concentration (≈110 g/L) in which the initial added quantity of nitrogen was the same as in the previous glucose-based experiments (*viz*. both peptone and yeast extract were added at 1.0 g/L). The initial C/N ratio in this experiment was ≈180 g/g. The total volume of the fermentation was 3.0 L, and the working volume was 2.0 L. The bioreactor was fitted with four probes and two six-bladed turbines. The culture vessel was inoculated with 100 mL (5.0% *v*/*v* inoculum) of exponential pre-culture (see composition of the pre-culture above). The incubation temperature was controlled automatically at 28 ± 1 °C. The agitation rate of the culture was adjusted to 350 ± 10 rpm. A cascade aeration (0.2–2.5 vvm) was employed to maintain the dissolved oxygen tension (DOT) at values ≥20% *v*/*v* of saturation, which indicates full aerobic conditions in the performed trial [30]. The pH was automatically controlled at the desired value by adding base quantities of 5 M KOH.

### 2.3. Analytical Methods

Cells from the whole content of the flasks (≈50 mL) or content of ≈20 mL from the bioreactor were separated from the medium through centrifugation (7690 × *g*, 4 °C, 15 min), while the supernatant was used for CA and sugars determination. The microbial biomass was collected, and it was repeatedly washed with deionized water and centrifuged until the supernatant was absolutely free of fermentation media. The dry cell weight (DCW) was calculated after drying at 85 °C for 48 h and cooling down in a desiccator. The dry biomass was homogenized using a mortar, and a solution of chloroform:methanol (2:1, *v*/*v*) [31] was added for microbial lipids extraction. The organic solvent was separated from the cell debris using filter paper, and it was evaporated under vacuum. The lipids were gravimetrically quantified in pre-weighed round bottom flasks.

The concentration of carbon sources was determined by high-performance liquid chromatography (Waters 600E Associatio, Milford, MA, USA) equipped with a UV (Waters 486, Milford, MA, USA) and an RI (Waters 410, Milford, MA, USA) detector. The column used was an Aminex HPX-87H (Biorad, Hercules, CA, USA) (300 mm length × 7.8 mm internal diameter). The mobile phase was a 5 mM H_2_SO_4_ aqueous solution with a 0.6 mL/min flow rate at 65 °C. Given that *iso*-citric was not sufficiently separated from CA, it was assayed, according to Papanikolaou et al. [32]. *Iso*-citric acid represented a quantity of 7–11% *w*/*w* of the sum of total citric acid (CA + *iso*-citric acid) produced. In all representations done in the present study, the concentration of CA was presented.

The concentration of free amino nitrogen in the fermentation medium was performed according to the method proposed by Filippousi et al. [30]. The analysis of ammonium ions in the growth medium was performed with an ammonium selective electrode (Hach 95–12), as proposed in Papanikolaou et al. [26].

The amount of phenolic compounds concentration was determined using the Folin–Ciocalteau method. The absorbance was measured at 750 nm wavelength, and it was expressed as gallic acid equivalents (GAE). The decolorization of the fermented media was performed according to Sarris et al. [14]. Briefly, the samples were diluted 30-fold, the pH value was adjusted around 6, and the absorbance was measured at 395 nm.

The determination of total IPs was performed based on the method described by Sarris et al. [14]. Briefly, 0.05 g of dry biomass was immersed in 10 mL of 2 M HCl, followed by a 30 min heat treatment (100 °C). The liquid phase was separated via filtration, and it was neutralized with 10 mL of 2 M NaOH.

The analysis of fatty acid composition in microbial oil was conducted with the production of fatty acid methyl esters (FAMEs) employing transesterification with sodium methoxide followed by esterification with methanol using HCl as a catalyst. The determination of FAMEs was carried out by a Fisons 8060 (Markham, ON, Canada) gas chromatography system equipped with a Chrompack (São Paulo, Brazil) column (60 m × 0.32 mm) and a flame ionization detector using helium as carrier gas (2.0 mL/min) [33].

### 2.4. Data Analysis

In all of the kinetic experiments performed, both in shake-flask and bioreactor trials, each experimental point presented in the figures and the tables was the mean value of two measurements deriving from two independent experiments, realized with different inocula. The SE in most of the experimental points was ≤15%. The data were plotted using Kaleidagraph 4.0 Version 2005, showing the mean values with the standard error mean.

### 2.5. Nomenclature

X (DCW in g/L)—Dry yeast cells; L (in g/L)—Intra-cellular lipids; IPs (in g/L)—Intra-cellular polysaccharides; CA (in g/L)—Citric acid; FAN (in mg/L)—free amino nitrogen; NH_4_^+^ (in mg/L)—ammonium ions; S (in g/L)—Glucose; Phen (in g/L)—Total phenolic compounds (GAE); Y_X/S_ (in g/g)—yield of biomass on glucose consumed (g of produced total dry weight per g of glucose assimilated); Y_L/X_ (in g/g)—Intra-cellular lipids in total biomass (g of lipids per g of DCW); Y_IPs/X_ (in g/g)—Intra-cellular polysaccharides in total biomass (g of polysaccharides per g of DCW); Y_CA/S_ (in g/g)—Yield of citric acid produced on consumed glucose (g of produced citric acid per g of glucose assimilated); Productivity (in g/L/h)—citric acid productivity. Indices 0, cons and max indicate the initial, consumed, and maximum values of each compound throughout fermentations.

## 3. Results

The three employed *Y. lipolytica* strains were tested on media composed of S_0_ ≈ 50 g/L under nitrogen-limited conditions (utilization of peptone and yeast extract at 1.0 g/L of each; initial C/N ratio employed ≈80 g/g) in order to favor the secretion of secondary metabolites (mostly citric acid) and potentially the accumulation of cellular lipids. The obtained results of the performed trials as regards biomass and lipid production of the screened strains are illustrated in Table 1.

Differences in the physiological patterns of the employed *Y. lipolytica* strains are depicted according to the obtained results; for instance, for the case of the microorganism ACA-YC 5033, this strain produced some lipid quantities (L = 0.7 g/L, simultaneous lipid in the DCW values ≈ 15% *w*/*w*) at the first growth phase (*viz*. the balanced phase, the duration of which as followed by the kinetics of FAN was up to 40–50 h after inoculation, and thereafter nitrogen-limited conditions occurred), and afterward, despite the non-negligible concentration of glucose that remained unconsumed into the medium, lipid in DCW values decreased. Throughout the culture phases, and irrespective of the presence (or absence) of quantities of assimilable nitrogen into the medium, IPs in the DCW values presented *grosso modo* similar values (up to 30–35% *w*/*w*). The strain LMBF Y-46 presented a relatively similar bio-kinetic pattern, although the lipid in the DCW values were slightly higher than the ones of the strain ACA-YC 5033, with polysaccharides in the DCW values remaining constant (in values of *c.* 32–35% *w*/*w*) during the culture (Table 1). A different pattern was observed for the strain LMBF Y-47, which maintained non-negligible lipid in the DCW values (28–30% *w*/*w*, L = 3.4–3.5 g/L) even at the late fermentation steps, with lipid values (in g/L and in g per g of total DCW) constantly increasing as the fermentation proceeded. On the other hand, all microorganisms at the nitrogen-limited growth phase secreted quantities of CA. *Y. lipolytica* ACA-YC 5033 produced the highest CA quantities among the three tested strains, with CA_max_ quantity being = 26.7 g/L (with the concomitant conversion yield of CA produced per unit of consumed glucose, Y_CA/S_, being ≈ 0.53 g/g) that had been achieved at the end of the culture. Noticeably lower CA_max_ quantities have been reported for the strain LMBF Y-46 (= 17.1 g/L, simultaneous Y_CA/S_ ≈ 0.36 g/g) and LMBF Y-47 (= 15.0 g/L, simultaneous Y_CA/S_ ≈ 0.31 g/g).

The fatty acid (FA) composition of the cellular lipids produced by the three *Y. lipolytica* strains is presented in Table 2 (analysis performed at the stationary growth phase of the cultures).

In agreement with most reports that appeared in the literature (see, i.e., state-of-the-art review articles such as those by Bellou et al. [34] and Valdés et al. [27]), the principal FAs found in variable quantities were mainly the oleic acid (C18:1) and the palmitic acid (C16:0).

Given that *Y. lipolytica* ACA-YC 5033 presented non-negligible production of CA in the conducted shake-flask experiments (see previous paragraph), it was decided to proceed with further cultivation of this strain on glucose-based experiments under higher S_0_ concentration in a bioreactor trial. Indeed, the strain ACA-YC 5033 was cultured on media with S_0_ ≈ 110 g/L, with the initial nitrogen concentration being added as peptone and yeast extract at a concentration of 1.0 g/L of each of the mentioned compounds (the same initial nitrogen quantity as in the previous shake-flask trials) and significant glucose assimilation and CA production occurred (see Figure 1a). The glucose assimilation rate was almost linear, as was the production rate of CA. The consumption rate of glucose was calculated by the formula $r_{S}=-\frac{\Delta S}{\Delta t}\;$ throughout the culture, was ≈0.61 g/L/h. 180 h after inoculation, the CA concentration was = 57.0 g/L, while for all experimental points, the concentration of *iso*-citric acid corresponded to 5–10% *w*/*w* of that of total citric acid (= CA + *iso*-citrate) produced. Typically, at the first culture steps (*viz*. at the balanced growth phase, that, as in shake-flask trials, occurred up to t ≈ 40 h), glucose was mainly consumed for total DCW production to occur. Thereafter, while biomass concentration remained *grosso modo* constant (X ranging between 7.0 and 8.0 g/L), the CA concentration drastically increased with a maximum quantity of 57.0 g/L being achieved at t = 180 h (the volumetric CA productivity was calculated by the formula $r_{CA}=\frac{\Delta CA}{\Delta t}\;$ and was ≈0.32 g/L/h). Although the produced CA was mostly non-growth-associated metabolite, the illustration of CA produced per remaining glucose into the fermentation medium for the whole set of data revealed quite accurately (*R* = 0.983) the global conversion yield of CA produced per unit of glucose consumed (Y_CA/S_), which was = 0.51 g/g (Figure 1b).

From the above-mentioned analysis, it can be seen that the strain *Y. lipolytica* ACA-YC 5033 was revealed to be a very interesting CA producer on media in which glucose (even at high initial concentrations) was added into the medium. In the next step, it was decided to perform trials in which glucose was added into OMW-based media to identify the possible effect of inhibiting compounds found in the waste towards the cells of the studied microorganism. Table 3 depicts the fermentation efficiency with the yeast *Y. lipolytica* ACA-YC 5033 cultivated on synthetic glucose-based media (control) as well as on OMW-based media at two Phen_0_ concentrations and a C/N ratio of ≈114 g/g. Glucose was added into the medium to yield a S_0_ concentration ≈ 40 g/L, typical to the OMWs deriving from three-phase centrifugation systems with low water consumption [21]. As can be observed, nitrogen-limited media favored the CA production. The increase in Phen_0_ significantly enhanced the CA production. Therefore, an interesting stimulatory effect of the added phenolic compounds upon the secretion of CA by *Y. lipolytica* was observed, at least for the moderate Phen_0_ concentrations found in the medium; the maximum CA concentration of 25.8 g/L (Y_CA/S_ = 0.63 g/g) was achieved at trials having ≈1.0 g/L of Phen_0_ concentration while the maximum Y_CA/S_ value obtained for the trials having ≈2.0 g/L of Phen_0_ was 0.54 g/g (see Table 3). On the other hand, the maximum biomass concentration of 7.9 g/L was noted at the blank fermentations (no OMW added).

As seen in Table 3, the accumulation of microbial oil was quite low (L_max_ up to 0.70 g/L corresponding to lipid in the DCW values ≤ 10% *w*/*w* for all performed trials). The increasing Phen_0_ concentration seemed to negatively affect the lipid concentration compared to the control fermentation (phenolic-free). The presence of ≈1.0 g/L Phen_0_ in the fermentation media did not affect the Y_L/X_ (up to 0.10 g/g), while a further increase resulted in decreased values of Y_L/X_ (Table 3). The Y_L/X_ values were much lower than 20% even though nitrogen-limited conditions prevailed in the growth media (OMW-based media; FAN = 5.52 ± 0.75 mg/L after 24 h and control medium; FAN = 10.11 ± 0.92 mg/L at 72 h) and glucose was consumed, suggesting a non-typical oleaginous behavior for the yeast *Y. lipolytica* (Table 3). This could be attributed to the shift of metabolism towards the production of IPs and enhanced secretion of CA into the medium (Table 3).

The IP production was interesting in all instances, verifying the assumption that the microbial metabolism shifted towards IP (and CA) formation and acted as a detriment to lipid biosynthesis. In all of the trials performed, the concentration of IPs ranged between 1.0 and 1.3 g/L, with simultaneous IPs in the DCW values ranging between *c*. 15.0 to *c*. 18.0% *w*/*w*, values significantly higher than the respective ones of lipid in DCW (see Table 3). It is also noted that the non-negligible IPs in the DCW values (up to 20% *w*/*w*) were observed at the early growth phase for all trials performed (0–50 h after inoculation), a period in which assimilable nitrogen was found in excess into the culture medium (data not presented), in some disagreement with the literature, suggesting that nitrogen-limited conditions are requested for the onset of polysaccharides (and cellular lipids) accumulation inside the yeast cells [27,30,33,34].

At the early stage of fermentation, the so-called “trophophase”, the catabolism of glucose, led mainly to biomass formation that reached 87.6%, 90.0% and 64.1% of its maximum value with respect to the control experiment in an OMW-based medium with 1.0 g/L and 2 g/L of Phen_0_, at 48 h of fermentation. The secondary anabolic activity of CA synthesis occurred when the yeast strain entered the so-called “idiophase”, which is related to nitrogen exhaustion from the culture media. More specifically, FAN was almost entirely depleted from the media after 24 h of fermentation, reaching 5.5 ± 0.8 mg/L in the case of OMW-based media. This event directly triggered CA biosynthesis that reached *c.* 6.5 g/L at 24 h (as mentioned already, at that time, the conditions in the medium were limited in nitrogen) and showed a high production rate (Figure 2a). CA production plateaued and did not show any significant variations after 120 h in the case of OMW-based media with 1.0 g/L Phen_0_ (Figure 2a) and after 168 h at 2.0 g/L Phen_0_, since glucose was not available in considerable amounts in the fermentation media (consumed by >97% in both experiments). This could indicate that the yeast cells passed to a stationary phase. Surprisingly, FAN was consumed after 72 h of fermentation, reaching 10.11 ± 0.92 mg/L at 72 h when the control media was investigated, while CA production was low at this point of fermentation (1.7 g/L). Although the produced CA was mostly non-growth-associated metabolite, the illustration of CA produced per remaining glucose into the fermentation medium for the whole set of data revealed quite accurately (*R* = 0.989) the global conversion yield of CA produced per unit of glucose consumed (Y_CA/S_) that was =0.66 g/g (Figure 2b).

The next set of fermentations focused on the effect of carbon-limited media (N-rich media) on CA, biomass, IPs and lipids production (Table 4). To achieve this, a C/N ratio of 16 g/g was applied via the addition of 4.0 g/L (NH_4_)_2_SO_4_ and 2.0 g/L yeast extract. The biomass and lipids production were significantly induced with increasing Phen_0_, with a maximum respective value of 25.3 g/L and 0.5 g/L when 1.0 g/L Phen_0_ was applied. The Y_X/S_, calculated by the formula $Y_{X/S}=\frac{X_{max}-X_{0}}{S_{cons}}$ was also favored in the presence of phenolics (Y_X/S_ ≈ 0.60 g/g; the respective value in the blank experiment, without OMWs added into the medium, was ≈0.38 g/g; see Table 4). The IP quantity was significantly higher (2.5–3.9 g/L) compared to the trial performed under nitrogen-limited conditions (see Table 3 and Table 4) with quite satisfying Y_IPs/X_ values (0.13–0.17 g/g). It should be mentioned that in this set of experiments, CA was not detected due to the low C/N ratio that was employed. Apparently, the yeast strain did not enter the idiophase since nitrogen was not completely exhausted from the growth media. The FAN was consumed by 60.12% at the end of fermentation (residual FAN = 38.4 ± 2.6 mg/L).

Comparing the two different C/N ratios, rationally, the lowest C/N ratio of ≈16 g/g led to significantly higher biomass formation (15.7–25.3 g/L) due to N great availability in the fermentation medium (Table 3 and Table 4). In this case, the Y_X/S_ was determined by ≈50.3–71.5% folds higher than values obtained at the nitrogen-limited trial (C/N ≈ 114 g/g).

Figure 3 presents the fermentation efficiency of *Y. lipolytica* ACA-YC 5033 when cultivated on OMW-based media at Phen_0_ ≈ 1.0 g/L and a C/N ratio equal to ≈16 g/g. The production of biomass in the carbon-limited experiments followed similar trends with the nitrogen-limited fermentations. The DCW production of *Y. lipolytica* ACA-YC 5033 reached 93.4%, 89.5% and 78.0% of its maximum value at 48 h in the control and OMW-based medium with 1.0 g/L (Figure 3a) and 2.0 g/L of Phen_0_, respectively. The glucose of the media was quickly and completely assimilated within the first *c.* 80 h of fermentation. This contrasted with the nitrogen-limited experiment where glucose was assimilated more gradually, with the maximum sugar assimilation measured after 120 h (Figure 2b). In most cases, the IP production was rapid up to 72 h, with slight variations thereafter. Similarly with the previous set of nitrogen-limited experiments, cellular lipid production and Y_L/X_ were negligible, confirming that the microbial metabolism has been shifted towards mostly DCW production and, to a lesser extent, IP synthesis in detriment to lipids biosynthesis, in accordance with most literature reports associated with the production of single-cell protein (for critical state-of-the-art review see Koukoumaki et al. [35]). Also, the presence of phenolic compounds in the control media did not seem to affect IP accumulation. Illustration of X produced per remaining glucose into the fermentation medium for the whole set of data revealed quite accurately (*R* = 0.983) the global conversion yield of X produced per unit of glucose consumed (Y_X/S_) that was =0.57 g/g (the obtained value was slightly lower than the one calculated from the fermentation point where the X_max_ concentration was achieved through the formula $Y_{X/S}=\frac{X_{max}-X_{0}}{S_{cons}}$; see Table 4) (Figure 3b).

The fatty acid profile of intra-cellular lipids that were produced under the different fermentation regimes is presented in Table 5. In all cases, oleic acid (C18:1) was the predominant fatty acid (>50%), followed by palmitic acid (C16:0), linoleic (C18:2) and palmitoleic acids (C16:1). The RSU fluctuated throughout the fermentations (0.1–0.4) while differences within same experiments were not significant. C16:1 and C18:0 did not exceed 10% in any of the examined cases, while alterations were monitored throughout fermentation without any pattern. The highest Phen_con_ (≈2 g/L) in both C/N ratios that were examined led to lipids rich in C18:1, with percentages that varied within 60.3–70.2%, while C18:2 was also found in substantial amounts. It can be observed that elevated amounts of C18:1 are mostly attributed to a significant decrease of C16:0 and C16:1.

The C18:2 content of lipids derived from the C/N ratio of 114 g/g at ≈2.0 g/L phenolics was equal to 18.3% at the early stage of fermentation (48 h); it decreased to 14.4% after 144 h, and finally, it showed a slightly increasing tendency (15.3%) at 192 h. When 1 g/L of phenolics was applied in the medium, C18:2 followed a decreasing tendency throughout fermentation, reaching from 14.4% at 24 h to 12.1% after 168 h of fermentation. Lipids derived from the control experiment (free of phenolics) contained much lower C18:2 amounts (10.5–11.5%).

In the case of the C/N ratio of 16 g/g (Phen_0_ ≈ 2 g/L), the highest C18:2 of 17.4% was monitored at 48 h while it was considerably decreased (14.3%) at 96 h. Enhanced amounts of C18:2 (17.2–17.3%) were also detected in lipids produced in synthetic media. The Phen_0_ of ≈1.0 g/L yielded lipids with a C18:2 percentage of 13.6–15.2% at the early stages of fermentation (24–48 h), while prolonged fermentation time resulted in a decrease (11.5%) of this value.

Figure 2c and Figure 3c present the removal of phenolic compounds and color achieved by *Y. lipolyica* ACA-YC 5033. Surprisingly enough, overall higher degrees of dephenolization and decolorization were observed in the media with a 114 g/g C/N ratio, besides the fact that a significantly lower amount of cell biomass was produced. It seems that the ability of the strain to remove phenolics and color from OMW was negatively affected by the lower C/N ratio. The removal of phenolics was observed after 24 h of fermentation, while after 50–70 h, no significant changes were observed. Maximum dephenolization (20.5% *w*/*w*) was achieved after 72 h in the nitrogen-limited media containing ≈ 1 g/L phenolics. The highest decolorization percentage of 46.2% was achieved in the fermentation of the nitrogen-limited media containing ≈ 2 g/L phenolics after 120 h.

Given that the achieved CA_max_ concentration in the fermentations of the OMW-based media was not significantly high (*viz*. CA_max_ = 25.8 g/L) as compared to the literature (even for wild-type *Y. lipolytica* strains, CA_max_ concentrations up to 40–45 g/L can be routinely achieved, whereas for mutant or genetically engineered strains, the corresponding concentrations can be in some cases ≥ 100 g/L; for critical review see Cavallo et al. [12]), it was decided to perform a supplementary shake-flask experiment in which higher S_0_ concentrations were added into the medium, with the initial nitrogen quantities remaining as previously (i.e., a combination of S_0_ adjusted to *c*. 100 g/L with 0.5 g/L of (NH_4_)_2_SO_4_ and 0.5 g/L of the yeast extract, with the initial C/N ≈ 286 g/g was chosen). In this trial, OMWs were added into the medium to yield a Phen_0_ ≈ 1.0 g/L that, as mentioned in the previous experiments, stimulated the production of CA (see Table 3). It should also be mentioned that high S_0_ concentrations similar to the ones of the supplementary experiment (i.e., glucose in quantities ≥ 80 g/L) can be found in OMWs originating from traditional press extraction systems [18,20,21]. The obtained results are illustrated in Figure 4a–c.

From the obtained results (see Figure 4a), it can be seen that in the first steps of the culture (i.e., the balanced growth phase that occurred up to the first 50 h of the culture), the microorganism consumed the available nitrogen quantity and also quantities of glucose, and mostly DCW production was observed. After *c.* 50 h, and when the concentration of the assimilable nitrogen virtually became limiting for microbial growth, biomass production reached a plateau (to *c.* 7.0–8.0 g/L), and the onset of CA secretion into the medium occurred. No further biomass increase was observed since restricted quantities of lipids (lipids in the DCW values ranging between 8.0 and 15% *w*/*w*) and moderate quantities of polysaccharides (polysaccharides in the DCW values ranging between 18.0 and 24.0% *w*/*w*) were obtained throughout the culture with no tendency of further increase (kinetics not shown). After nitrogen depletion from the medium (as mentioned, this happened *c*. 50 h after inoculation), CA in significant quantities was secreted, while the assimilation rate of glucose was uninterrupted despite the nitrogen limitation imposed (Figure 4a).

The kinetics for the whole fermentation (Figure 4b) demonstrated that for all of the culture period, glucose assimilation rate, calculated as previously by the formula $r_{S}=-\frac{\Delta S}{\Delta t}$ throughout the culture, was constant (≈0.28 g/L/h), while the very interesting CA_max_ concentration of 60.2 g/L was obtained *c*. 380 h after inoculation (simultaneous volumetric productivity ≈ 0.16 g/L/h). Finally, an illustration of the CA produced per remaining glucose into the fermentation medium for the whole set of data accurately revealed (*R* > 0.99) the global conversion yield of CA produced per unit of glucose consumed (Y_CA/S_) that presented a quite satisfactory value of 0.66 g/g (Figure 4c). Finally, as in all the previously performed fermentations on OMW-based media, the production of DCW and CA was accompanied by a satisfactory phenolic compounds’ removal (up to 19.5% *w*/*w*) and a noticeable decolorization (up to 47.0%) of the fermentation medium (kinetics not presented).

## 4. Discussion

The generation of large quantities of OMWs in short periods of time poses important environmental risks for the Mediterranean basin. The OMW accounts for 50% of the waste generated by olive oil production, and it is estimated that 30 million tons are produced every year in the Mediterranean. Due to the presence of recalcitrant compounds, these residues are difficult to be treated [35,36,37]. In relatively recent experimental works that have appeared in the literature, OMWs have been employed to “dilute” “concentrated” residues and waste streams in a zero-waste and water-saving approach. Therefore, “concentrated” residues like molasses and crude glycerol have been diluted [14,24,25], while strains of the yeast *Y. lipolytica* have been revealed as very capable candidates, presenting significant resistance upon the recalcitrant compounds found in various types of OMWs [14,24,25,38].

CA has been reported as one of the most intensively produced acids with numerous applications in the food sector, pharmaceuticals, cosmetics, biopolymer industry and medicine for the formulation of drug delivery systems and tissue engineering. Many fungal and bacterial strains have been investigated for CA biosynthesis. CA commercialization is challenging since it is produced as an intermediate of the energy metabolic pathway, and its secretion occurs when specific nutrient imbalances, i.e., nitrogen exhaustion, prevail in the fermentative micro-environment. Although several microorganisms can produce CA, *Y. lipolytica* has emerged as a promising alternative due to its high yield, low-cost production, and ease of genetic manipulation. Moreover, *Y. lipolytica* is an avirulent yeast species that historically has been used in the food and chemical industries for its ability to grow on agro-industrial wastes and by-products and synthesize various compounds such as organic acids, sugar alcohols, and single-cell protein [32,35,38,39,40]. Several strains of *Y. lipolytica* were studied in glucose-based media to investigate their ability to accumulate cellular lipids, which led to the observation that nitrogen-limited conditions resulted in CA production [41]. Citric acid is the immediate precursor of cellular lipid formation in oleaginous microorganisms, meaning that their metabolism can be directed to produce biomass, lipids or CA by adjusting medium components and genetic engineering [42]. The effective bioconversion of renewable feedstocks into CA has been achieved with wild-type, recombinant or engineered strains of *Y. lipolytica* [12,40,43]. According to the literature, CA production by wild-type strains of *Y. lipolytica* can be, in rare cases, in concentrations up to 100 g/L CA, whereas mutant strains may generate more than 100 g/L, while maximum CA concentrations up to 160 g/L have been indicated in some reports in the literature [5,12,44,45,46,47,48,49,50,51,52,53,54,55,56].

In the present study, the maximum CA production obtained (CA = 57.0 g/L with Y_CA/S_ = 0.51 g/g in bioreactors with glucose used as substrate; CA = 60.2 g/L with Y_CA/S_ = 0.66 g/g in flasks with glucose/OMWs blends used as substrate) can be considered as quite satisfactory, specifically considering that a wild-type strain was employed as cell factory and “waste” materials (specifically in the case of OMWs/glucose blends) were used as carbon sources. As mentioned, when wild-type strains are implicated in the CA production process, values of some tenths g/L are considered as very satisfactory, whereas by mutant (mostly acetate- or aconitase-ones) or genetically engineered strains, the corresponding concentrations can often be ≥100 g/L. Likewise, the maximum values of conversion yields of CA produced per unit of substrate (mostly sugar or glycerol) consumed can be within the range of the values achieved in the current submission (0.40–0.70 g/g; rarely values ≥ 0.80 g/g are achieved) [12,40]. As far as the wild-type *Y. lipolytica* strains are considered, very interesting values (up to 62 g/L with very high volumetric productivity of ≈1.4 g/L/h) have been obtained by the strain H222, cultured on glucose in fed-batch bioreactor systems [48]. For the same strain, impressive quantities of CA (i.e., up to 98.3 g/L) have been reported in repeated fed-batch biosensor-controlled substrate-feeding fermentation of 5–7 days cycle time, when growth has been performed on glucose, with concomitant Y_CA/S_ obtained = 0.69 g/g [49]. Moreover, very high total citric acid concentrations (i.e., CA + *iso*-citric acid = 101.4 g/L) have been reported by the strain LMBF Y-46 cultivated on crude glycerol in highly agitated and aerated bioreactor trials [50]. It was indeed paradoxical that this strain, as shown in the current investigation, did not proceed to high citric acid production during its cultures on glucose. On the other hand, it must be indicated that significantly different culture conditions were implicated in the present study as compared to the previous investigation (use of shake flasks instead of highly agitated bioreactor, different carbon source, etc.). In any case, it is noted that these two mentioned studies in which the wild-type LMBF Y-46 and H222 strains have been used [49,50] are amongst the very few ones in which wild *Y. lipolytica* strains have been employed as microbial cell factories for the production of CA, and concentrations near 100 g/L have been reported to be achieved. On the other hand, there is a number of reports in which mutants on genetically modified strains have attained notably higher CA_max_ quantities compared with the current submission (i.e., CA quantities ranging between 102–140 g/L [9,51,52,53], whereas, as stated in the previous paragraphs, the highest CA concentration reported so far in the literature was that of 200 g/L [13].

The fact that *Y. lipolytica* is able to catabolize a variety of hydrophilic and hydrophobic carbon sources has motivated researchers to investigate CA production efficiency in low- or negative-cost agro-industrial side streams [54], while in many cases the utilization of “waste” materials into the growth medium (like the OMWs) resulted in lower CA_max_ quantities obtained compared to the current study (CA_max_ = 60.2 g/L, Y_CA/S_ = 0.66 g/g in flasks with glucose/OMWs used as substrate; see Figure 4b,c). For instance, the evaluation of cane molasses, glucose syrup and fructose syrup as an alternative feedstock for *Y. lipolytica*, led to CA production of up to 38 g/L, which was further improved in subsequent bioreactor experiments [5]. Pretreated straw cellulose, waste cooking oil, pineapple waste, and crustacean waste have also been successfully used as carbon sources to produce CA and other metabolites in *Y. lipolytica* fermentations [57,58,59,60], with CA concentrations achieved being lower than the ones obtained in blends of OMWs and glucose. In contrast, higher quantities compared to the present study (CA up to 100–140 g/L) were produced by *Y. lipolytica* in a 10-L bioreactor with defined nutrient media enriched with rapeseed oil or aspen waste [9]. In another case, whey and grape supplemented with glucose or fructose led to the production of 49.2 g/L and 32.1 g/L of CA, respectively, demonstrating, as the present investigation, that *Y. lipolytica* cultures can efficiently valorize a variety of agricultural wastes [61]. Arslan et al. [62] developed a novel non-aseptic culture process and demonstrated the potential of a cold-adapted lactose-positive *Y. lipolytica* strain to effectively produce satisfying amounts of CA (33.3 g/L). Finally, as mentioned in the previous paragraphs, crude glycerol, the main by-product deriving from biodiesel production facilities, has been revealed as an excellent substrate amenable to be converted into CA by (mostly acetate-negative or genetically modified mutants of) the yeast *Y. lipolytica*, with achieved CA production presenting in many cases higher final concentrations than the present study; for instance, the mutant strains Wratislavia AWG7 and A-101-1.22 have been revealed capable of producing the significantly high concentrations of CA ranging between 119.1 and 160.5 g/L, during growth on this renewable resource in fed-batch or repeated batch bioreactor experiments [63,64]. However, crude glycerol cannot be considered a “waste” material *per se* (in most cases this substrate has a small acquisition cost contrary to the OMWs that present a negative value [26,31,32,35]), while, as mentioned, in most cases, the high CA final concentrations previously indicated (i.e., CA ≥ 100 g/L) have been achieved with the aid of mutant and not wild-type strains [9,53,63,64].

Total cellular lipids, both in relative and absolute values, were produced in quite low amounts in all the examined cases (C/N ratios and Phen_0_). In fact, the specific strain of *Y. lipolytica* ACA-YC 5033 that was investigated in this study demonstrated a non-typical oleaginous profile with total intra-cellular lipid contents being much lower than 20%. Microorganisms can be characterized as oleaginous when they are able to accumulate more than 20% of lipids in their total cellular dry weight (Y_L/X_ > 0.2 g/g) [27,34,65]. The oleaginous profile of *Y. lipolytica* strains is mostly strain-dependent, while the fermentation mode, i.e., batch, fed-batch, or continuous strategies, contributes to a lower extent. The intra-cellular lipids that were produced by *Y. lipolytica* ACA-YC 5033 were characterized with respect to their fatty acid composition. Analyses were performed throughout the fermentation to monitor any alterations that could possibly occur (Table 5). Overall, the fatty acid profile of cellular lipids was typical of *Y. lipolytica* strains, with C18:1 being the principal fatty acid, followed by C16:0. The sum of C16:1 and C18:0 was lower than 20% in all cases, while C18:2 was detected in substantial amounts. 

In the study of Papanikolaou et al. [50], *Y. lipolytica* strain LMBF-Y46 was reported to produce lipids with a much lower content of C18:1 (28.4–43.0%), in agreement with the present study, where trials were performed with this strain on glucose. Moreover, the strain LMBF-Y46 produced small amounts of lauric acid (0.2–3.9%), myristic acid (<2.5%), arachidic acid (0.5–5.9%) and behenic acid (1.5–3.0%) that were also detected when the yeast strain was cultivated on batch shake flask or fed-batch bioreactor trials using glycerol as the fermentation media. The most interesting observation was the high unsaturation degree of cellular lipids. The unsaturated fraction of fatty acids was 70–89% of total fatty acids, and it was mainly attributed to the presence of C18:1 and C18:2 (Table 5). Similar to this result, higher concentrations of the cellular FAs C16:0 and C18:0 have been noted in the beginning of other batch cultures performed by several *Y. lipolytica* strains on glucose or glycerol. These concentrations decreased as fermentations proceeded. This could be attributed to the lower DOT values in these cultures at the early growth steps of the balanced growth phase of *Y. lipolytica* yeast [50].

The low quantities of cellular lipids produced in this study seem to be related to IP synthesis. Although there is no evidence for an interconversion pathway of lipids and polysaccharides, glycogen synthesis is a competing pathway to lipid synthesis [66,67]. In contrast to CA production, the substrate with a C/N ratio of 16 g/g somehow favored the IP content of *Y. lipolytica,* resulting in a 1.65-fold increase, compared to the substrate with C/N ratio of 118 g/g, although according to the theory, the biosynthesis of intra-cellular polysaccharides requires culture conditions limited in nitrogen [27,34,68]. The accumulation of IPs in the cells of oleaginous fungi starts at the early trophophase and peaks after nitrogen exhaustion during the stationary phase, which was also observed in this study [69]. In contrast, non-oleaginous fungi produce appreciable IP quantities even in nitrogen-abundant media [70,71]. High IP synthesis during the stationary phase and basically unaltered IP content during the oleaginous phase is a physiological characteristic shared by *Y. lipolytica* and other oleaginous yeasts *C. curvatus* and *R. toruloides* [72]. The maximum absolute IP quantities produced by *Y. lipolytica* in the present study (3.9 g/L) compare favorably with those reported for *Metschnikowia* sp. (1.7–7.7 g/L), *Rhodosporidium toruloides* (2.5–3.7 g/L), and *Rhodotorula* sp. (1.1–2.4 g/L) cultivated in biodiesel-derived glycerol and glycerol/xylose blends [73]. Tzirita et al. [25] studied how the production of metabolic compounds is affected by different salt concentrations in media containing OMW and glycerol and observed a low IP production (1.5 g/L) but high polyol, lipid and CA production (12.9, 2.5, and 32.0 g/L respectively).

There are only a few publications demonstrating dephenolization by yeasts in substrates, such as OMW. *Y. lipolytica* cultured in OMW enriched with biodiesel-derived glycerol achieved significant dephenolization (40–62%) and noticeable decolorization [25]. In a study comparing sterilized and unsterilized glucose-based substrates with added OMW, 51% and 58% of the phenolics and color were removed in both cases, while a favorable effect of the OMW additions on lipid accumulation was observed. Crude glycerol and OMW mixed in different ratios were utilized in nitrogen-limited shake-flask cultures of *Y. lipolytica,* and approximately 30% and 10% decolorization and dephenolization were determined [14]. Similarly, 31–26% color removal was achieved by two strains of *Y. lipolytica* in substrates with different levels of OMW [17]. The removal of phenolics (20.5% *w*/*w*) achieved in this study is lower than that reported in existing literature, but decolorization (46.2%) compares favorably with the reported values. The ability of white-rot fungi to produce extra-cellular enzymes such as laccases, manganese and lignin peroxidases, which can effectively degrade phenolic compounds, is absent in non-GMO yeasts [21,74]. It has been speculated that the phenolic and color removal observed in substrates such as OMW during yeast fermentation is due to partial utilization and potential adsorption of phenolic compounds within the yeast cells [14].

## 5. Conclusions/Future Perspectives

Three novel, wild-type *Y. lipolytica* strains were cultured under nitrogen-limited conditions on glucose-based media in order to identify their potential for CA biosynthesis and production of other valuable metabolites (i.e., intra-cellular lipids and polysaccharides). In accordance with most literature reports, all studied strains showed an untypical oleaginous character; therefore, the cellular lipid in the DCW values remained at a level ≤30.0% *w*/*w* while simultaneous CA production was noted. Also, variable quantities of intra-cellular polysaccharides were produced. In the strain, ACA-YC 5033, interesting CA quantities were synthesized. This strain cultivated in stirred tank bioreactor experiments, with high glucose concentrations (S_0_ ≈ 110 g/L), produced CA up to 57 g/L (simultaneous yield on sugar consumed Y_CA/S_ ≈ 0.52 g/g). The same strain bioremediated OMWs containing variable quantities of glucose. It was demonstrated that the presence of phenolic compounds in concentrations found in typical OMWs stimulated the production of CA in this strain. Under nitrogen-limited conditions in which OMWs had partially replaced tap water and significant glucose concentrations had been loaded (S_0_ ≈ 100 g/L, initial phenolic compounds [Phen_0_] ≈ 1.0 g/L; this is a composition similar to that of OMWs generated from the traditional press extraction systems) the notable CA quantity of 60.2 g/L, with simultaneous Y_CA/S_ = 0.66 g/g, was obtained. Carbon-limited conditions with Phen_0_ ≈ 1.0 g/L favored the production of yeast biomass (up to 25.3 g/L). Irrespective of the carbon or nitrogen-limited conditions employed, satisfactory phenolic compound removal and waste decolorization were observed simultaneously with the production of biomass and metabolites. Through all the mentioned analysis, it can be suggested that medium-to-high glucose content phenol-containing effluents (i.e., OMWs deriving from three-phase centrifugation systems with low water consumption or OMWs deriving from the traditional press extraction systems) can be successfully converted into valuable compounds with the use of *Y. lipolytica* ACA-YC 5033. Finally, low-organic loaded phenol-containing wastewaters (i.e., three-phase OMWs or some types of table olive processing wastewaters) can successfully substitute tap water in order to “dilute” “concentrated” residues and waste streams in a zero-waste and water-saving approach in fermentation bioprocesses involving *Y. lipolytica* ACA-YC 5033.

## Figures and Tables

**Figure 1 microorganisms-11-02243-f001:**
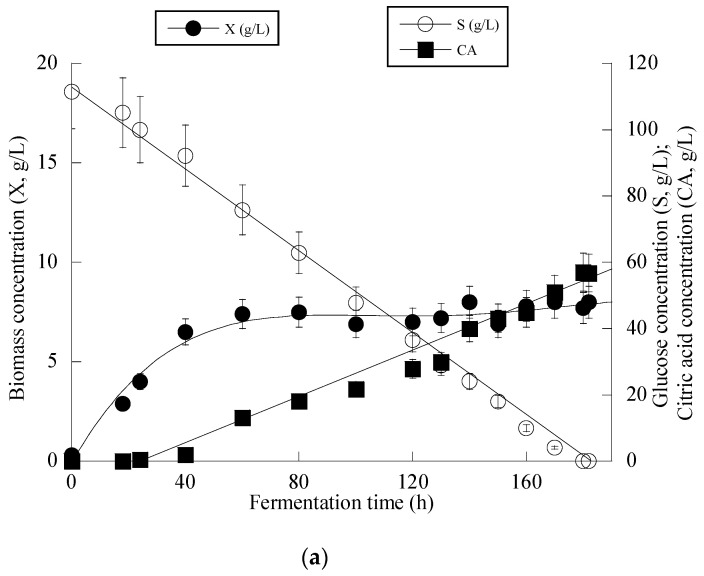
Evolution of biomass (X) (●), citric acid (CA) (■) and glucose (S) (○) (**a**) and citric acid (CA) (■) vs. glucose (**b**) during batch bioreactor fermentation of *Yarrowia lipolytica* ACA-YC 5033 on synthetic glucose-based media.

**Figure 2 microorganisms-11-02243-f002:**
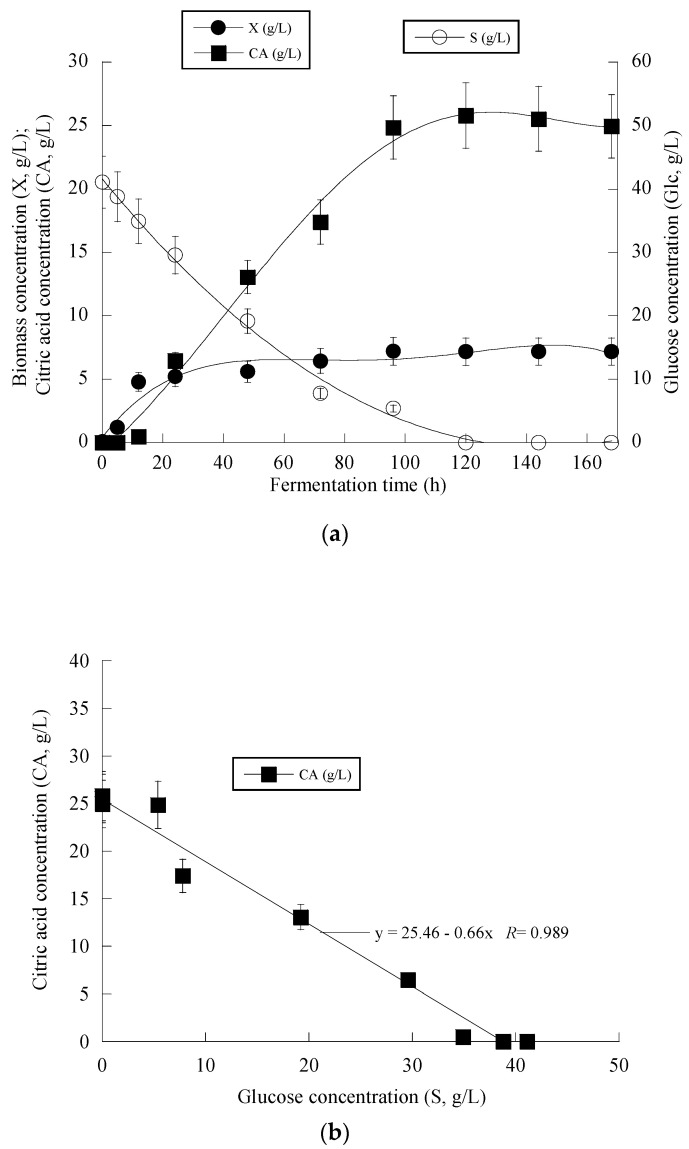
Evolution of biomass (X) (●), citric acid (CA) (■) and glucose (S) (○) (**a**), citric acid (CA) (■) vs. glucose (**b**) and decolorization (♦) and phenol removal (□) (**c**) during batch flask fermentation of *Y. lipolytica* ACA-YC 5033 on media composed of OMWs and glucose, under nitrogen-limited conditions. Initial phenolic compounds into the OMW-based media ≈ 1.0 g/L.

**Figure 3 microorganisms-11-02243-f003:**
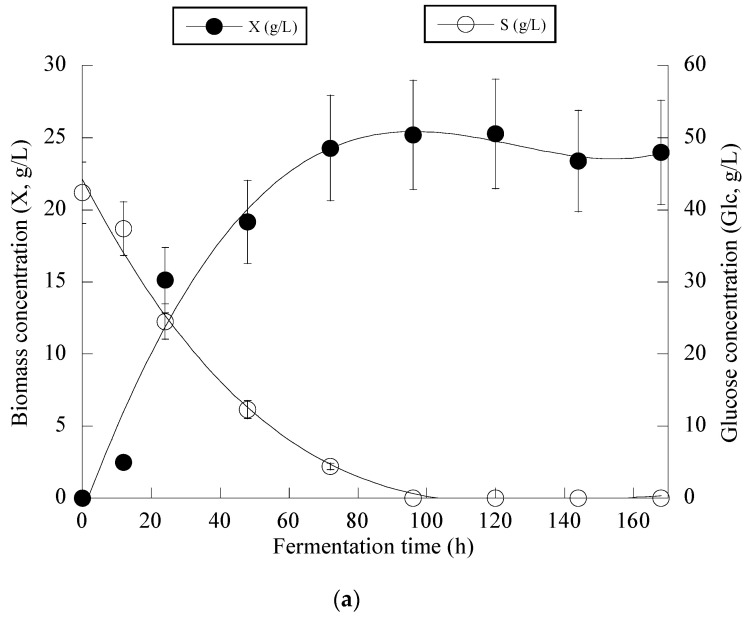
Evolution of biomass (X) (●) and glucose (S) (○) (**a**), biomass (X) (●) vs. glucose (**b**) and decolorization (♦) and phenol removal (□) (**c**) during batch flask fermentation of *Y. lipolytica* ACA-YC 5033 on media composed of OMWs and glucose, under carbon-limited conditions. Initial phenolic compounds into the OMW-based media ≈ 1.0 g/L.

**Figure 4 microorganisms-11-02243-f004:**
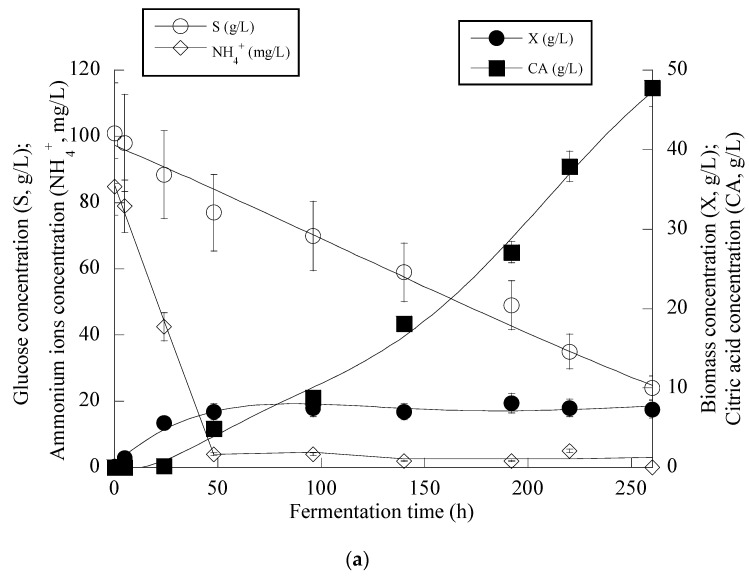
Evolution of biomass (X) (●), citric acid (CA) (■), glucose (S) (○) and ammonium ions (NH4+) (◊) at the relatively early growth steps (up to 250 h) (**a**), biomass (X) (●), citric acid (CA) (■) and glucose (S) (○) for the whole fermentation period (**b**) and citric acid (CA) (■) vs. glucose (**c**) during batch flask fermentation of *Y. lipolytica* ACA-YC 5033 on media composed of OMWs and glucose, under nitrogen-limited conditions. Initial phenolic compounds into the OMW-based media ≈ 1.0 g/L, S_0_ ≈ 100 g/L.

**Table 1 microorganisms-11-02243-t001:** Experimental results originated from the kinetics of *Y. lipolytica* strains grown on commercial glucose in shake-flask experiments.

Strain	Point	Time (h)	S_cons_ (g/L)	X (g/L)	L (g/L)	IPs (g/L)	Y_IPs/X_ (g/g)	Y_L/X_ (g/g)
ACA-YC 5033	a	24	6.1 ± 1.7	4.8 ± 0.9	0.7 ± 0.2	1.6 ± 0.7	0.34	0.15
	b	195	50.4 ± 1.1	7.1 ± 1.4	0.6 ± 0.2	2.2 ± 0.5	0.31	0.08
LMBF Y-46	a	120	42.9 ± 2.4	9.5 ± 1.8	2.5 ± 0.4	3.3 ± 0.4	0.35	0.26
	b	170	47.1 ± 2.1	11.4 ± 1.9	1.8 ± 0.4	3.9 ± 0.9	0.34	0.16
LMBF Y-47	a	152	44.1 ± 2.1	11.4 ± 1.8	3.4 ± 1.0	2.9 ± 0.8	0.25	0.30
	b	172	48.2 ± 2.3	12.4 ± 2.0	3.5 ± 0.9	2.7 ± 0.8	0.22	0.28

Representations of fermentation time (h), biomass (X, g/L), lipid (L, g/L), glucose consumed (S_cons_, g/L), lipid in dry biomass (Y_L/X;_ g/g), polysaccharides (IPs, g/L) and polysaccharides in the DCW (Y_IPs/X_; g/g) when the maximum quantity of lipids in dry cell weight (Y_L/X_; g/g) (a) and the maximum concentration of biomass (X, g/L) (b) was obtained. Each experimental point is the mean value of two determinations (SE < 15%).

**Table 2 microorganisms-11-02243-t002:** Fatty acid composition of the cellular lipids produced by *Y. lipolytica* strains cultivated on glucose in shake-flask experiments (S_0_ ≈ 50 g/L). The time of fermentation for the determination of the fatty acid composition was between 150 and 200 h after inoculation. Each experimental point is the mean value of two determinations (SE < 15%).

Yeast Strain	C16:0	C18:0	C18:1	C18:2
*Y. lipolytica* ACA-YC 5033	16.1 ± 1.9	4.5 ± 0.6	53.1 ± 3.6	12.7 ± 2.4
*Y. lipolytica* LMBF Y-46	11.1 ± 1.9	5.0 ± 1.6	41.2 ± 2.9	22.4 ± 1.9
*Y. lipolytica* LMBF Y-47	25.9 ± 2.9	6.4 ± 1.3	46.9 ± 2.5	15.1 ± 2.9

**Table 3 microorganisms-11-02243-t003:** Fermentation efficiency of the yeast *Y. lipolytica* ACA-YC 5033 on commercial glucose-based media (control) as well as on olive-mill-wastewater-based media at two initial phenolic concentrations (Phen_con_) under shake-flasks nitrogen-limited conditions.

	Phen_0_ (g/L)	Time (h)	X (g/L)	L (g/L)	IPs (g/L)	CA (g/L)	Y_X/S_ (g/g)	Y_L/X_ (g/g)	Y_IPs/X_ (g/g)	Y_CA/S_ (g/g)	Productivity (g/L/h)
a	0	120	7.5 ± 0.2	0.7 ± 0.1	1.2 ± 0.2	11.8 ± 0.1	0.21	0.09	0.16	0.33	
b	0	168	7.9 ± 0.4	0.3 ± 0.1	1.2 ± 0.1	18.9 ± 0.9	0.20	0.04	0.15	0.49	0.113
a	≈1.0	96	7.2 ± 0.4	0.7 ± 0.1	1.2 ± 0.2	24.9 ± 0.8	0.20	0.10	0.16	0.70	
b	≈1.0	120	7.2 ± 0.6	0.2 ± 0.1	1.3 ± 0.1	25.8 ± 0.9	0.18	0.03	0.18	0.63	0.215
a	≈2.0	120	7.3 ± 0.5	0.4 ± 0.1	1.3 ± 0.1	12.3 ± 0.5	0.22	0.05	0.18	0.37	
b	≈2.0	192	6.2 ± 0.6	0.4 ± 0.1	1.0 ± 0.2	20.8 ± 1.1	0.16	0.06	0.16	0.54	0.108

Nitrogen was added into the medium as (NH_4_)_2_SO_4_ at 0.5 g/L and yeast extract at 0.5 g/L to achieve a C/N molar ratio equal to ≈ 114 g/g. Fermentation data are presented when the maximum biomass (X) and lipids (L) (a), as well as citric acid (CA) (b), were achieved throughout each fermentation.

**Table 4 microorganisms-11-02243-t004:** Fermentation efficiency of the yeast *Y. lipolytica* ACA-YC 5033 on commercial glucose-based media (control) as well as on olive-mill-wastewaters-based media presenting two initial phenolics concentrations ≈1.0 g/L and 2.0 g/L (Phen_0_), in shake-flask carbon-limited experiments.

Phen_0_ (g/L)	Time (h)	X (g/L)	L (g/L)	IPs (g/L)	Y_X/S_ (g/g)	Y_L/X_ (g/g)	Y_IPs/X_ (g/g)
0	144	15.7 ± 0.5	0.3 ± 0.1	2.5 ± 0.8	0.38	0.02	0.16
≈1.0	120	25.3 ± 0.8	0.5 ± 0.1	3.3 ± 0.1	0.60	0.02	0.13
≈2.0	96	22.9 ± 0.9	0.4 ± 0.1	3.9 ± 0.2	0.60	0.02	0.17

Nitrogen was added into the medium as (NH_4_)_2_SO_4_ at 4.0 g/L and yeast extract at 2.0 g/L to achieve a C/N ratio equal to ≈16 g/g. Fermentation data are presented when the maximum biomass (X) was achieved throughout each fermentation.

**Table 5 microorganisms-11-02243-t005:** Fatty acid composition (%, *w*/*w*) and the ratio of saturated to unsaturated fatty acids (RSU) of microbial oil derived from *Y. lipolytica* ACA-YC 5033 cultivated in a glucose-based media (blank) and glucose-based media blended with olive mill wastewater (≈1.0 g/L and ≈2.0 g/L of phenolic compounds), under C/N molar ratios of ≈114 g/g and ≈16 g/g.

			Fatty Acid Methyl Esters (% *w*/*w*)					
	Phen_0_ (g/L)	Time (h)	C16:0	C16:1	C18:0	C18:1	C18:2	RSU
**C/N ≈ 114**	0	48	14.8 ± 0.3	9.9 ± 0.2	7.8 ± 0.2	56.8 ± 0.8	10.7 ± 0.2	0.3
		72	18.6 ± 0.5	7.2 ± 0.3	7.8 ± 0.2	54.9 ± 0.7	11.5 ± 0.4	0.4
		144	14.5 ± 0.6	10.0 ± 0.4	4.7 ± 0.2	60.2 ± 0.4	10.5 ± 0.1	0.2
	≈1.0	48	14.8 ± 0.2	8.0 ± 0.2	4.9 ± 0.1	57.9 ± 0.3	14.4 ± 0.4	0.2
		72	17.3 ± 0.5	10.0 ± 0.4	4.2 ± 0.1	54.1 ± 0.5	14.3 ± 0.4	0.3
		120	15.7 ± 0.5	8.1 ± 0.3	5.1 ± 0.4	57.4 ± 0.8	13.7 ± 0.1	0.3
		168	21.2 ± 0.3	6.4 ± 0.2	7.5 ± 0.3	52.8 ± 0.4	12.1 ± 0.4	0.4
	≈2.0	48	11.5 ± 0.2	3.0 ± 0.1	4.3 ± 0.2	62.9 ± 0.5	18.3 ± 0.4	0.2
		96	12.3 ± 0.1	4.3 ± 0.1	5.1 ± 0.2	63.7 ± 0.8	14.6 ± 0.3	0.2
		144	12.1 ± 0.3	3.4 ± 0.3	3.9 ± 0.1	66.2 ± 0.5	14.4 ± 0.5	0.2
		192	14.7 ± 0.2	4.6 ± 0.1	5.1 ± 0.3	60.3 ± 0.3	15.3 ± 0.4	0.2
**C/N ≈ 16**	0	24	16.6 ± 0.4	8.6 ± 0.2	5.0 ± 0.2	52.6 ± 0.2	17.2 ± 0.4	0.3
		144	15.9 ± 0.2	8.9 ± 0.4	5.0 ± 0.2	52.9 ± 0.2	17.3 ± 0.2	0.3
	≈1.0	24	21.5 ± 0.2	2.0 ± 0.1	9.2 ± 0.2	53.7 ± 0.5	13.6 ± 0.5	0.4
		48	18.0 ± 0.4	8.0 ± 0.2	8.2 ± 0.3	50.6 ± 0.7	15.2 ± 0.2	0.4
		144	19.8 ± 0.4	9.8 ± 0.2	10.0 ± 0.3	48.9 ± 0.5	11.5 ± 0.4	0.4
	≈2.0	24	10.0 ± 0.4	3.5 ± 0.1	2.6 ± 0.1	70.2 ± 0.7	11.6 ± 0.4	0.1
		48	8.2 ± 0.1	4.2 ± 0.1	2.3 ± 0.1	67.8 ± 0.8	17.4 ± 0.3	0.1
		96	17.0 ± 0.3	2.3 ± 0.1	4.9 ± 0.1	61.4 ± 0.5	14.3 ± 0.4	0.3

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
