# Peer review of "Growth Response of Non-Conventional Yeasts on Sugar-Rich Media: Part 2: Citric Acid Production and Circular-Oriented Valorization of Glucose-Enriched Olive Mill Wastewaters Using Novel Yarrowia lipolytica Strains"

_microorganisms, 2023, doi:10.3390/microorganisms11092243_

Round 1
Reviewer 1 Report
Greetings, Editor thank you for providing me with the opportunity to review the article. I reviewed the article with the title ``Growth response of non-conventional yeasts on sugar-rich me- 2 dia: Part 2. Citric acid production and circular-oriented valori- 3 zation of glucose-enriched olive mill wastewater using novel 4 Yarrowia lipolytica strains``. Overall, the article is well written and explained in term of scientific terms. I have a few suggestions as given here. Further, I am recommending a minor corrections.
1. The title should be changed to more suitable. The present seems like a sentence. Kindly present a scientific title. Take this comment as a serious point. It’s very weird in the present form.
2. The abstract seems to be a bit ok. Please add one more introductory line of your objective in beginning of abstract. Highlight the core finding.
3. Keywords is fine.
4. Research gap should be delivered in a clearer way with directed necessity for the future research work.
5. The methodology part also seems ok. But please check the abbreviation. Please revise your paper accordingly since several abbreviation issue occurs on several spots in the paper.
6. The overall article results, and its discussion seems fine to me.
7. Regarding the replications, authors confirmed that replications of experiment were carried out. However, these results are not shown in the manuscript, how many replicated were carried out by experiment?
8. Section 5 should be renamed Conclusion and Future perspectives. Conclusion section is missing some perspective related to the future research work, quantify main research findings, highlight relevance of the work with respect to the field aspect.
9. To avoid grammar and linguistic mistakes, major level English language should be thoroughly checked. Please revise your paper accordingly since several language issue occurs on several spots in the paper.
To avoid grammar and linguistic mistakes, major level English language should be thoroughly checked. Please revise your paper accordingly since several language issue occurs on several spots in the paper.
Author Response
Reference: 2519784
Title: Growth response of non-conventional yeasts on sugar-rich media: Part 2: Citric acid production and circular-oriented valorization of glucose-enriched olive mill wastewaters using novel Yarrowia lipolytica strains
Dear Editorial Office of “Microorganisms”,
You will find in the following pages the comments of the referees and our responses. In the revised manuscript, all of our changes and additions are found in yellow fond. To more properly respond to referee 2 (“… more precisely, in most of the results and discussions they state that the results are worse than those reported in the literature”), we have performed a supplementary experiment (see pages 16/26 and 17/26) in which we have obtained a noticeable citric acid (CA) production during growth on blends of olive-mill wastewaters and glucose (CA=60.2 g/L, global conversion yield of CA produced per unit of glucose consumed YCA/S=0.66 g/g) that is amongst the most interesting CA production values when wastes are employed as microbial substrates of Yarrowia lipolytica. Our responses to the comments of referees follow in the nest pages.
Sincerely,
Professor Seraphim Papanikolaou
Referee 1
Greetings, Editor thank you for providing me with the opportunity to review the article. I reviewed the article with the title “Growth response of non-conventional yeasts on sugar-rich media: Part 2. Citric acid production and circular-oriented valorization of glucose-enriched olive mill wastewater using novel Yarrowia lipolytica strains”. Overall, the article is well written and explained in term of scientific terms. I have a few suggestions as given here. Further, I am recommending a minor corrections.
- The title should be changed to more suitable. The present seems like a sentence. Kindly present a scientific title. Take this comment as a serious point. It’s very weird in the present form.
Response: Thank you for your comment. Unfortunately, we cannot change our title, since already the “Part 1” has been published in “Microorganisms” and already is found in databases (like “Scopus”). The title is as follows: “Growth response of non-conventional yeasts on sugar-rich media: Part 2: Citric acid production and circular-oriented valorization of glucose-enriched olive mill wastewaters using novel Yarrowia lipolytica strains”. Alternatively, and if the referee is not satisfied, we could maintain the sentence “Growth response of non-conventional yeasts on sugar-rich media: Part 2:”, and we could discuss potential other titles concerning our manuscript.
- The abstract seems to be a bit ok. Please add one more introductory line of your objective in beginning of abstract. Highlight the core finding.
Response: Thank you for your comment. We have changed our “Abstract” accordingly.
- Keywords is fine.
Response: Thank you for your comment.
- Research gap should be delivered in a clearer way with directed necessity for the future research work.
Response: Thank you for your comment. Done. Additions were performed in both the “Introduction” and the “Conclusions / Future perspectives”.
- The methodology part also seems ok. But please check the abbreviation. Please revise your paper accordingly since several abbreviation issue occurs on several spots in the paper.
Response: Thank you for your comment. Done. We have tried to do our best concerning this comment.
- The overall article results, and its discussion seems fine to me.
Response: Thank you for your comment.
- Regarding the replications, authors confirmed that replications of experiment were carried out. However, these results are not shown in the manuscript, how many replicated were carried out by experiment?
Response: Thank you for your comment. Please see section “2.4. Data analysis”.
- Section 5 should be renamed Conclusion and Future perspectives. Conclusion section is missing some perspective related to the future research work, quantify main research findings, highlight relevance of the work with respect to the field aspect.
Response: Thank you for your comment. Done. We have tried to do our best concerning this comment.
- To avoid grammar and linguistic mistakes, major level English language should be thoroughly checked. Please revise your paper accordingly since several language issue occurs on several spots in the paper.
Response: Thank you for your comment. Done. We have tried to do our best concerning this comment.
Referee 2
- A very extensive work has been presented in which it is difficult to find a focus. The title refers to the recycling of wastewater from olive mills and the production of citric acid. However, the work loses its focus. It also describes the process of CA synthesis on glucose, accumulation of lipids and polysaccharides, fatty acid profile, etc. Considering this, the discussion is also difficult to follow, especially considering the numerous comparisons with, for example, glycerol and other waste materials on which YL grows.
Response: Thank you for your comment. We do not consider that our manuscript was difficult to follow. In any case, we have performed additions in both the “Introduction” and the “Discussion” sections, to facilitate the reader. We consider that discussions and comparisons with glycerol should remain, since this is a low-cost material, and many works in the literature have been carried out concerning CA production with this substrate.
- Moreover, the authors give numerous results and report on the production of CA and do not provide any new results; more precisely, in most of the results and discussions they state that the results are worse than those reported in the literature. The paper needs to be rewritten with a clear reporting focus.
Response: Thank you for your comment. The approach that was proposed in the current manuscript (simultaneous valorization of olive mill wastewater-based media and production of metabolic compounds like CA, lipids and yeast biomass) we consider that it is quite original, although the production of CA was lower that the maximum values appeared in the literature. CA production in the glucose-based bioreactor experiment (57 g/L, YCA/S≈0.52 g/g), was quite satisfactory, considering that our strain was a new wild-type and not a genetically engineered or a mutant one. In any case, we have performed a supplementary experiment (see pages 16/26 and 17/26) in which we have obtained a noticeable citric acid (CA) production during growth on blends of olive-mill wastewaters and glucose (CA=60.2 g/L, YCA/S=0.66 g/g) that is amongst the highest CA production values when wastes are employed as microbial substrates of Yarrowia lipolytica, as indicated in our discussion.
Referee 3
The authors made the requested corrections.
Response: Thank you for your comment.
Referee 4
The present manuscript is of interest to the field and fit the scope of the journal. However, it should be improved further.
- The figures should be improved. Especially, the symbols are too big to be concordant with the figures, please revise throughout the manuscript.
Response: Thank you for your comment. The symbols were changed accordingly.
- The presentation in the manuscript should be simplified. For the discussion part, the first paragraph is much too long and should be better put in the background. Or it should be significantly shortened.
Response: Thank you for your comment. Done. We have tried to do our best concerning this comment.
- Conclusion should be take-home message, discussion and reference should be avoided.
Response: Thank you for your comment. Done. We have tried to do our best concerning this comment.
Reviewer 2 Report
A very extensive work has been presented in which it is difficult to find a focus. The title refers to the recycling of wastewater from olive mills and the production of citric acid. However, the work loses its focus. It also describes the process of CA synthesis on glucose, accumulation of lipids and polysaccharides, fatty acid profile, etc. Considering this, the discussion is also difficult to follow, especially considering the numerous comparisons with, for example, glycerol and other waste materials on which YL grows.
Moreover, the authors give numerous results and report on the production of CA and do not provide any new results; more precisely, in most of the results and discussions they state that the results are worse than those reported in the literature. The paper needs to be rewritten with a clear reporting focus.
Author Response

(The authors gave the same response as above.)

Reviewer 3 Report
The authors made the requested corrections.
Author Response

(The authors gave the same response as above.)

Reviewer 4 Report
The present manuscript is of interest to the field and fit the scope of the journal. However, it should be improved further.
1. the figures should be improved. Especially, the symbols are too big to be concordant with the figures, please revise throughout the manuscript.
2. The presentation in the manuscript should be simplified. For the discussion part, the first paragraph is much too long and should be better put in the background. Or it should be significantly shortened.
3. Conclusion should be take-home message, discussion and reference should be avoided.
Author Response

(The authors gave the same response as above.)
